# FINDING AND ONLY FINDING LOCAL NASH EQUILIBRIA BY BOTH PRETENDING TO BE A FOLLOWER

## ABSTRACT

Finding Nash equilibria in two-player differentiable games is a classical problem in game theory with important relevance in machine learning. We propose double Follow-the-Ridge (double-FTR), an algorithm that locally converges to and only to local Nash equilibria in general-sum two-player differentiable games. To our knowledge, double-FTR is the first algorithm with such guarantees for general-sum games. Furthermore, we show that by varying its preconditioner, double-FTR leads to a broader family of algorithms with the same convergence guarantee. In addition, double-FTR avoids oscillation near equilibria due to the real-eigenvalues of its Jacobian at fixed points. Empirically, we validate the double-FTR algorithm on a range of simple zero-sum and general sum games, as well as simple Generative Adversarial Network (GAN) tasks.

## 1 INTRODUCTION

Much of the recent success in deep learning can be attributed to the effectiveness of gradient-based optimization. It is well-known that for a minimization problem, with appropriate choice of learning rates, gradient descent has convergence guarantee to local minima (Lee et al., 2016, 2019). Based on this foundational result, an array of accelerated and higher-order methods have since been proposed and widely applied in training neural networks (Duchi et al., 2011; Kingma and Ba, 2014; Reddi et al., 2018; Zhang et al., 2019b).

However, once we leave the realm of minimization problems and consider the multi-agent setting, the optimization landscape becomes much more complicated. Multi-agent optimization problems arise in diverse fields such as robotics, economics and machine learning (Foerster et al., 2016; Von Neumann and Morgenstern, 2007; Goodfellow et al., 2014; Ben-Tal and Nemirovski, 2002; Gemp et al., 2020; Anil et al., 2021).

A classical abstraction that is especially relevant for machine learning is two-player differentiable games, where the objective is to find global or local Nash equilibria. The equivalent of gradient descent in such a game would be each agent applying gradient descent to minimize their own objective function. However, in stark contrast with gradient descent in solving minimization problems, this gradient-descent-style algorithm may converge to spurious critical points that are not local Nash equilibria, and in the general-sum game case, local Nash equilibria might not even be stable critical points for this algorithm (Mazumdar et al., 2020b)!

These negative results have driven a surge of recent interest in developing other gradient-based algorithms for finding Nash equilibria in differentiable games. Among them is Mazumdar et al. (2019), who proposed an update algorithm whose attracting critical points are only local Nash equilibria in the special case of zero-sum games. However, to the best of our knowledge, such guarantees have not been extended to general-sum games.

We propose double Follow-the-Ridge (double-FTR), a gradient-based algorithm for general-sum differentiable games that locally converges to and only to differential Nash equilibria. Double-FTR is closely related to the Follow-the-Ridge (FTR) algorithm for Stackelberg games (Wang et al., 2019), which converges to and only to local Stackelberg equilibria (Fiez et al., 2019). Double-FTR can be viewed as its counterpart for simultaneous games, where each player adopts the "follower" strategy in FTR.

The rest of this paper is organized as follows. In Section 2, we give background on two-player differentiable games and equilibrium concepts. We also explain the issues with using gradient-descent-style algorithm on such games. In Section 3, we present the double-FTR algorithm and prove its local convergence to and only to differential Nash equilibria. We also identify a more general class of algorithms that share these properties. We discuss recent works directly relevant to double-FTR in Section 4 and other related work in Section 5. In Section 6, we show empirical evidence of double-FTR's convergence to and only to local Nash equilibria.

## 2 BACKGROUND

### 2.1 TWO-PLAYER DIFFERENTIABLE GAMES AND EQUILIBRIUM CONCEPTS

In a general-sum two-player differentiable game, player 1 aims to minimize $f : \mathbb{R}^{n+m} \to \mathbb{R}$ with respect to $\boldsymbol{x} \in \mathbb{R}^n$, whereas player 2 aims to maximize $g : \mathbb{R}^{n+m} \to \mathbb{R}$ with respect to $\boldsymbol{y} \in \mathbb{R}^m$. Following the notation in Mazumdar et al. (2019), we denote such the game as $\{(f, -g), \mathbb{R}^{n+m}\}$. We also make the following assumption on the twice-differentiability of $f$ and $g$.

**Assumption 1.** $\forall\, \boldsymbol{x} \in \mathbb{R}^n, \boldsymbol{y} \in \mathbb{R}^m$, $f$ and $g$ are twice-differentiable, and the second derivatives are continuous. Also, $\nabla_{\boldsymbol{xx}}^2 f$ and $\nabla_{\boldsymbol{yy}}^2 g$ are invertible.

For two rational, non-cooperative players, their optimal outcome is to achieve a local Nash equilibrium (Ratliff et al., 2013). A point $(\boldsymbol{x}^*, \boldsymbol{y}^*)$ is a local Nash equilibrium[1] of $\{(f, -g), \mathbb{R}^{n+m}\}$ if there exists open sets $\mathcal{S}_{\boldsymbol{x}} \subset \mathbb{R}^n, \mathcal{S}_{\boldsymbol{y}} \subset \mathbb{R}^m$ such that $\boldsymbol{x}^* \in \mathcal{S}_{\boldsymbol{x}}, \boldsymbol{y}^* \in \mathcal{S}_{\boldsymbol{y}}$, and

$$f(\boldsymbol{x}^*, \boldsymbol{y}^*) \le f(\boldsymbol{x}, \boldsymbol{y}^*), \;\; g(\boldsymbol{x}^*, \boldsymbol{y}^*) \ge g(\boldsymbol{x}^*, \boldsymbol{y}), \;\; \forall \boldsymbol{x} \in \mathcal{S}_{\boldsymbol{x}}, \;\; \forall \boldsymbol{y} \in \mathcal{S}_{\boldsymbol{y}}.$$

A closely related notion of equilibrium is the differential Nash equilibrium (DNE) (Ratliff et al., 2013), which satisfies a second-order sufficient condition for local Nash equilibrium.

**Definition 2.1** (Differential Nash equilibrium). $(\boldsymbol{x}^*, \boldsymbol{y}^*)$ is a differential Nash equilibrium of $\{(f, -g), \mathbb{R}^{n+m}\}$ if the following two conditions hold:

- $\nabla_{\boldsymbol{x}} f(\boldsymbol{x}^*, \boldsymbol{y}^*) = 0$ and $\nabla_{\boldsymbol{y}} g(\boldsymbol{x}^*, \boldsymbol{y}^*) = 0$.

- $\nabla_{\boldsymbol{xx}}^2 f(\boldsymbol{x}^*, \boldsymbol{y}^*) \succ 0$ and $\nabla_{\boldsymbol{yy}}^2 g(\boldsymbol{x}^*, \boldsymbol{y}^*) \prec 0$.

The conditions of DNE are slightly stronger than that of local Nash equilibria in that the second-order conditions are definite instead of semi-definite. In this paper, we focus on DNE, as they make up almost all local Nash equilibria in the mathematical sense, and are well-suited for the analysis of second-order algorithms.

### 2.2 ISSUES WITH GRADIENT-BASED ALGORITHMS

A natural strategy for agents to search for local Nash equilibria in a differentiable game is to use gradient-based algorithms. The simplest gradient-based algorithm is the gradient descent-ascent (GDA) (Ryu and Boyd, 2016; Zhang et al., 2021b) (Algorithm 1) or its variants (Zhang et al., 2021a; Korpelevich, 1976; Mokhtari et al., 2020).

---

**Algorithm 1** Gradient descent-ascent (GDA)

---

**Require:** Number of iterations $T$, learning rate $\gamma$
1: **for** $t = 1, \ldots, T$ **do**
2:     $\boldsymbol{x}_{t+1} = \boldsymbol{x}_t - \gamma \nabla_{\boldsymbol{x}} f(\boldsymbol{x}_t, \boldsymbol{y}_t)$
3:     $\boldsymbol{y}_{t+1} = \boldsymbol{y}_t + \gamma \nabla_{\boldsymbol{y}} g(\boldsymbol{x}_t, \boldsymbol{y}_t)$
4: **end for**

---

Let $\boldsymbol{z} = \begin{bmatrix} \boldsymbol{x} \\ \boldsymbol{y} \end{bmatrix}$ and $\gamma > 0$ be the learning rate, a gradient-based update algorithm can be written as:

$$\boldsymbol{z}_{t+1} = \boldsymbol{z}_t - \gamma \boldsymbol{\omega}(\boldsymbol{z}_t). \tag{1}$$

---

[1]Note that local Nash equilibrium is not guaranteed to exist in nonconvex-nonconcave games ((Jin et al., 2020), Proposition 6), although the (non-)existence of local NE is out of the scope of this paper.

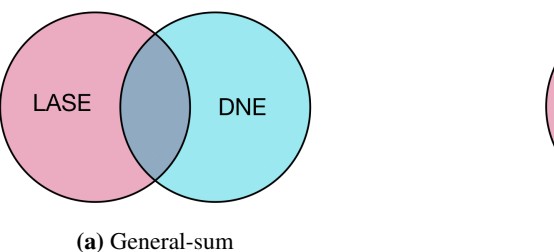

**(a)** General-sum         **(b)** Zero-sum

**Figure 1:** Venn diagrams showing the relationship between the set of locally asymptotically stable equilibria (LASE) of the GDA flow and the set of differential Nash equilibria (DNE) in two-player differentiable games. Note that for general-sum games, there exist DNE that are *unstable* for GDA flow.

The Jacobian of $\boldsymbol{\omega}(\boldsymbol{z})$ is defined as $\boldsymbol{J}(\boldsymbol{z}) := \frac{\partial \boldsymbol{\omega}(\boldsymbol{z})}{\partial \boldsymbol{z}}$. In the case of GDA, we have:

$$\boldsymbol{\omega}_{\text{GDA}}(\boldsymbol{z}) = \begin{bmatrix} \nabla_{\boldsymbol{x}} f(\boldsymbol{x}, \boldsymbol{y}) \\ -\nabla_{\boldsymbol{y}} g(\boldsymbol{x}, \boldsymbol{y}) \end{bmatrix}, \quad \boldsymbol{J}_{\text{GDA}} = \begin{bmatrix} \nabla^2_{\boldsymbol{xx}} f & \nabla^2_{\boldsymbol{xy}} f \\ -\nabla^2_{\boldsymbol{yx}} g & -\nabla^2_{\boldsymbol{yy}} g \end{bmatrix}.$$

Using the Jacobian matrix, we characterize the fixed points of equation 1.

**Definition 2.2** ((Strictly) stable fixed point). $\boldsymbol{z}^*$ is a stable fixed point of the discrete-time dynamical system in equation 1 if

$$\boldsymbol{\omega}(\boldsymbol{z}^*) = \boldsymbol{0} \quad \text{and} \quad \rho(\boldsymbol{I} - \gamma \boldsymbol{J}(\boldsymbol{z}^*)) \leq 1,$$

where $\rho(\cdot)$ denotes the spectral radius of a matrix. If we additionally have $\rho(\boldsymbol{I} - \gamma \boldsymbol{J}(\boldsymbol{z}^*)) < 1$, then $\boldsymbol{z}^*$ is a strictly stable fixed point.

Strictly stable fixed points are important for analysis, as they are locally asymptotically convergent (Galor, 2007), i.e. there exists an open set $\mathcal{S}_{\boldsymbol{z}}$ such that $\boldsymbol{z}^* \in \mathcal{S}_{\boldsymbol{z}}$ and $\lim_{t \to \infty} \boldsymbol{z}_t = \boldsymbol{z}^* \; \forall \boldsymbol{z}_0 \in \mathcal{S}_{\boldsymbol{z}}$.

A closely related concept is the locally asymptotically stable equilibrium (LASE) for the continuous-time system $\dot{\boldsymbol{z}} = -\boldsymbol{\omega}(\boldsymbol{z})$. (Ratliff et al., 2013).

**Definition 2.3** (Locally asymptotically stable equilibrium (LASE)). $\boldsymbol{z}^*$ is a locally asymptotically stable equilibrium of the continuous-time dynamics $\dot{\boldsymbol{z}} = -\boldsymbol{\omega}(\boldsymbol{z})$ if

$$\boldsymbol{\omega}(\boldsymbol{z}^*) = \boldsymbol{0} \quad \text{and} \quad \text{Re}(\lambda) > 0 \text{ for } \forall \lambda \in \text{spec}(\boldsymbol{J}(\boldsymbol{z}^*)),$$

where $\text{Re}(\cdot)$ denotes the real part of a complex number, and $\text{spec}(\cdot)$ returns the spectrum (i.e. the set of eigenvalues) of a matrix.

Note that when $\gamma \to 0$, strictly stable fixed points of equation 1 are equivalent to LASE of $\dot{\boldsymbol{z}} = -\boldsymbol{\omega}(\boldsymbol{z})$. In this paper, we prove convergence results in discrete-time (using Definition 2.2), but we often provide intuition using continuous-time concepts such as LASE.

Unfortunately, GDA is not guaranteed to converge to DNE, nor are DNE necessarily (strictly) stable fixed points of the GDA dynamics. Even in the special case of zero-sum games ($g = f$), GDA dynamics can still have stable fixed points that are not DNE (Daskalakis and Panageas, 2018; Mazumdar et al., 2020b). The relationship is shown in the Venn diagrams in Figure 1 (to eliminate the effect of $\gamma$, we show illustration in the continuous-time limit $\dot{\boldsymbol{z}} = -\boldsymbol{\omega}_{\text{GDA}}(\boldsymbol{z})$).

In Figure 2, we demonstrate the failure modes of GDA in zero-sum games. In 2a, GDA converges to a spurious strictly stable fixed point which is not DNE. In 2b, GDA fails to converge to the unique DNE (Hsieh et al., 2020). Instead, it goes into a limit cycle, due to the strong rotation introduced by large complex parts in its Jacobian eigenvalues. We stress that these pathologies are not limited to GDA, but common for many other first-order algorithms (Wang et al., 2019).

## 3   DOUBLE FOLLOW-THE-RIDGE

We propose double Follow-the-Ridge (double-FTR), an update rule for general-sum differential games that locally converges to and only to differential Nash equilibria. The double-FTR update is shown in Algorithm 2 (the arguments $\boldsymbol{x}_t$, $\boldsymbol{y}_t$ of $f$ and $g$ are dropped to avoid notational clutter).

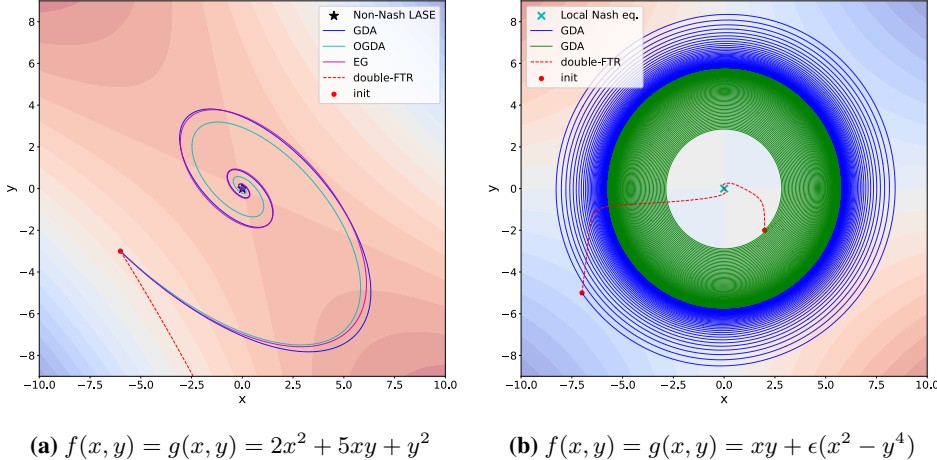

**(a)** $f(x, y) = g(x, y) = 2x^2 + 5xy + y^2$      **(b)** $f(x, y) = g(x, y) = xy + \epsilon(x^2 - y^4)$

**Figure 2:** Two examples of GDA failure modes in finding Nash equilibrium in zero-sum games. (a) GDA converges to the spurious strictly stable fixed point $(0,0)$, which is not a Nash equilibrium. Other first-order methods such as the optimistic GDA (OGDA) and extragradient (EG) converge to the spurious fixed point as well. (b) Instead of the unique Nash equilibrium $(0,0)$, GDA converges to a limit cycle both when initialized "inside" (green) and "outside" (blue). We use $\epsilon = 0.0001, \gamma = 0.01$.

---

**Algorithm 2** Double Follow-the-Ridge

---

**Require:** Learning rate $\eta_{\boldsymbol{x}}$ and $\eta_{\boldsymbol{y}}$; number of iterations $T$.
1: **for** $t = 1, \dots, T$ **do**
2:      $\boldsymbol{x}_{t+1} \leftarrow \boldsymbol{x}_t - \eta_{\boldsymbol{x}} \nabla_{\boldsymbol{x}} f - \eta_{\boldsymbol{y}} (\nabla^2_{\boldsymbol{xx}} f)^{-1} \nabla^2_{\boldsymbol{xy}} g \nabla_{\boldsymbol{y}} g$
3:      $\boldsymbol{y}_{t+1} \leftarrow \boldsymbol{y}_t + \eta_{\boldsymbol{y}} \nabla_{\boldsymbol{y}} g + \eta_{\boldsymbol{x}} (\nabla^2_{\boldsymbol{yy}} g)^{-1} \nabla^2_{\boldsymbol{yx}} f \nabla_{\boldsymbol{x}} f$
4: **end for**

---

Let $\boldsymbol{z} = \begin{bmatrix} \boldsymbol{x} \\ \boldsymbol{y} \end{bmatrix}$, $\gamma = \eta_{\boldsymbol{x}}$ and $c = \frac{\eta_{\boldsymbol{y}}}{\eta_{\boldsymbol{x}}}$, we can express Algorithm 2 in vectorized form (equation 2). To simplify the notation, we drop the subscript $t$ for $f$ and $g$.

$$\boldsymbol{z}_{t+1} = \boldsymbol{z}_t - \gamma \boldsymbol{\omega}_{\mathrm{FTR}}(\boldsymbol{z}_t), \quad \boldsymbol{\omega}_{\mathrm{FTR}}(\boldsymbol{z}_t) = \begin{bmatrix} \boldsymbol{I} & -(\nabla^2_{\boldsymbol{xx}} f)^{-1} \nabla^2_{\boldsymbol{xy}} g \\ -(\nabla^2_{\boldsymbol{yy}} g)^{-1} \nabla^2_{\boldsymbol{yx}} f & \boldsymbol{I} \end{bmatrix} \begin{bmatrix} \nabla_{\boldsymbol{x}} f \\ -c \nabla_{\boldsymbol{y}} g \end{bmatrix}.$$
(2)

### 3.1 LOCAL CONVERGENCE OF DOUBLE-FTR

In this section, we give our main theoretical result. First, we introduce an additional assumption.

**Assumption 2.** At fixed points of equation 2, $\boldsymbol{J}_{\mathrm{GDA}}(\boldsymbol{z})$ has full rank.

Assumption 2 ensures that in double-FTR, the additional terms in the update do not exactly cancel out the GDA terms. In practice, $\ell_2$ regularization might need to be added to the objective functions. Note a similar assumption is introduced in Mazumdar et al. (2019) Theorem 4.

Our main theoretical result is stated below.

**Theorem 1.** *Under Assumptions 1 and 2 and with an appropriate choice of learning rate $\gamma$, $\boldsymbol{z}^*$ is a strictly stable fixed point of the double-FTR update (equation 2) if and only if it is a differential Nash equilibrium of the game $\{(f, -g), \mathbb{R}^{n+m}\}$. Furthermore, at fixed points of equation 2, all eigenvalues of the Jacobian $\boldsymbol{J}_{\mathrm{FTR}} := \frac{\partial \boldsymbol{\omega}_{\mathrm{FTR}}}{\partial \boldsymbol{z}}$ are real.*

Intuitively, the first part of the theorem classifies the strictly stable fixed points of double-FTR, and the second part ensures that there is no rotation caused by complex eigenvalues in the neighbourhood of the DNEs. We defer the proof of Theorem 1 to Appendix A.

**Corollary 1** (Local convergence). *Let $\boldsymbol{z}^*$ be a DNE of the game $\{(f, -g), \mathbb{R}^{n+m}\}$. Under Assumptions 1 and 2 and with an appropriate choice of learning rate $\gamma$, there exists an open set $\mathcal{S}_{\boldsymbol{z}} \subset \mathbb{R}^{n+m}$ where $\boldsymbol{z}^* \in \mathcal{S}_{\boldsymbol{z}}$, such that when following equation 2, $\forall \boldsymbol{z}_0 \in \mathcal{S}_{\boldsymbol{z}}, \lim_{t \to \infty} \boldsymbol{z}_t \to \boldsymbol{z}^*$.*

*Proof.* The proof follows naturally by combining Theorem 1 with the local convergence of strictly stable fixed points (Galor (2007), Proposition 1.9). □

To the best of our knowledge, double FTR is the first algorithm with such local convergence result for general-sum games.

## 3.2 GENERAL PRECONDITIONERS

In the following remark, we show that double-FTR can be generalized to include a whole family of algorithms.

**Remark 1.** Theorem 1 applies to a more general version of the double FTR algorithm. In particular, we can generalize equation 2 to allow a broader class of "preconditioners":

$$z_{t+1} = z_t - \gamma \tilde{\omega}_{\text{FTR}}(z_t), \ \ \tilde{\omega}_{\text{FTR}}(z) = \begin{bmatrix} P_x & \\ & -P_y \end{bmatrix} J_{\text{GDA}}^\top(z_t) \begin{bmatrix} \nabla_x f \\ -c\nabla_y g \end{bmatrix}, \tag{3}$$

where $P_x$, $P_y$ are functions of $x, y$ respectively, which satisfy $P_x \succ 0 \iff \nabla_{xx}^2 f \succ 0$ and $P_y \prec 0 \iff \nabla_{yy}^2 g \prec 0$.

Equation 2 corresponds to the special case of $P_x = (\nabla_{xx}^2 f)^{-1}$, $P_y = (\nabla_{yy}^2 g)^{-1}$. The proof for Theorem 1 directly applies to the case of general preconditioners in Remark 1.

Remark 1 provides intuition on the convergence properties of double-FTR. Without the preconditioner $P_x$ and $P_y$, double-FTR reduces to Hamiltonian gradient descent (Mescheder et al., 2017; Balduzzi et al., 2018; Loizou et al., 2020; Abernethy et al., 2021), which is not guaranteed to only converge to local Nash equilibria. It is the introduction of the preconditioner that enables strictly stable fixed points to satisfy the second-order condition of DNE.

Remark 1 also sheds lights on how to derive a more practical algorithm. Naively implementing Algorithm 2 might cause instability when $\nabla_{xx}^2 f$ and $\nabla_{yy}^2 g$ are near singular. In practice, we use $(\nabla_{xx}^2 f \nabla_{xx}^2 f + \lambda I)^{-1} \nabla_{xx}^2 f$ instead of $(\nabla_{xx}^2 f)^{-1}$ in Algorithm 2 (where a small $\lambda > 0$ is the damping parameter). Note that this also allows us to drop the assumption on the invertibility of $\nabla_{xx}^2 f$ and $\nabla_{yy}^2 g$ in Assumption 1.

## 4 CONNECTION WITH OTHER ALGORITHMS

Mazumdar et al. (2019) proposed local symplectic surgery (LSS) – a gradient-based algorithm whose LASE are exactly local Nash equilibria in two-player zero-sum games. LSS avoids oscillatory behaviour at local Nash equilibria (similar to double-FTR). Compared to LSS, double-FTR appears to have a simpler form and enables a broader family of algorithms with such local convergence result in general-sum games.

The Follow-the-Ridge (FTR) algorithm (Wang et al., 2019) is closely related to our proposed double-FTR. FTR was proposed for two-player sequential games and is guaranteed to converge to and only to local minimax. FTR applies a gradient correction term on the follower in a sequential game, so that the agents approximately follow a ridge in the landscape of the objective function. The double-FTR can be viewed as a counterpart of FTR for simultaneous games. The update rule of double-FTR resembles that of FTR, with the gradient modification term applied on both players.

Another related algorithm is the Hamiltonian gradient descent (HGD) (Mescheder et al., 2017; Balduzzi et al., 2018; Loizou et al., 2020; Abernethy et al., 2021). HGD performs gradient-descent on the Hamiltonian, or the squared norm of the gradient. HGD is guaranteed to converge, as it is essentially a minimization problem. However, in general it is not guaranteed to converge only to local Nash equilibria. Interestingly, our double-FTR can be viewed as a preconditioned HGD.

## 5 RELATED WORK

Mazumdar et al. (2020b) introduced a general framework for competitive gradient-based learning. They characterized local Nash equilibria in terms of the critical points of the gradient algorithms.

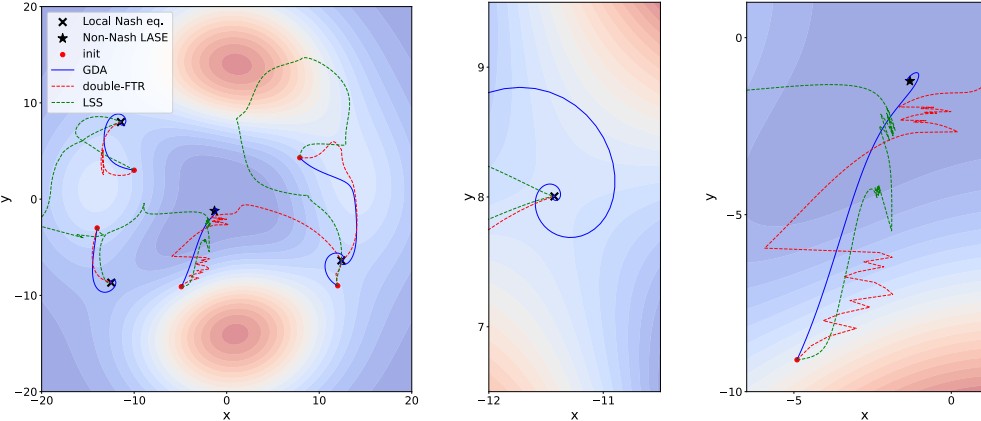

**Figure 3:** Left: evolution of GDA and double-FTR in the 2-D toy example from multiple initial points. Middle: zoom-in near a local Nash equilibrium point. Right: zoom-in near a non-Nash LASE for the GDA algorithm.

They showed the lack of convergence of the gradient algorithm in games, which motivated the development of the double-FTR algorithm.

Much work has focused on improving the dynamics in finding stable fixed points, which is crucial in applications such as GANs, where oscillation caused by eigenvalues with zero real parts of large imaginary parts in the gradient Jacobian can lead to training instability. Mescheder et al. (2017) proposes Consensus Optimization, which encourages agreement between the two players by introducing a regularization term in the objectives of both players. The regularization term results in a more negative real-part for the eigenvalues of the gradient Jacobian, therefore reduces oscillation and allows larger learning rates. Balduzzi et al. (2018); Gemp and Mahadevan (2018) proposes Symplectic Gradient Adjustment (SGA), which decomposes the gradient Jacobian into symmetric (potential) and asymmetric (Hamiltonian) parts and adds a gradient adjustment term for rapid convergence to stable fixed points. Schäfer and Anandkumar (2019) proposes Competitive Gradient Descent (CGD), whose update is given by the Nash equilibrium of a regularized bilinear approximation of the original game. Compared to other methods, CGD has the advantage of not needing to adapt step size when the interaction strength changes between players. Many other methods have been proposed with different strategies for predicting other agents' moves, such as Learning with Opponent Learning Awareness (LOLA) (Foerster et al., 2016) and optimistic gradient descent-ascent (OGDA) (Popov, 1980; Rakhlin and Sridharan, 2013; Daskalakis et al., 2018; Mertikopoulos et al., 2018). However, none of these existing methods address the problem of spurious (i.e. non-Nash) stable fixed points.

## 6 EXPERIMENTS

We conduct simple experiments to demonstrate the implications of our theoretical results. In Section 6.1, we show that the double-FTR algorithm empirically converges to and only to differential Nash equilibria, as predicted by Theorem 1. In Section 6.2, we show that double-FTR is able to converge to local Nash equilibria that naive gradient-play avoids in *general-sum* linear quadratic games. In Section 6.3, we demonstrate the practical implications of another property of double-FTR — eigenvalues of $\boldsymbol{J}_{\text{FTR}}$ at fixed points are real.

### 6.1 2-D TOY EXAMPLE

We consider the zero-sum game $\{f, -f\}, \mathbb{R}^2$ with the following 2-D function (same as in Mazumdar et al. (2019)):

$$f(x, y) = e^{-0.01(x^2 + y^2)}\big((0.3x^2 + y)^2 + (0.5y^2 + x)^2\big).$$

This function has several strictly stable fixed points for the GDA dynamics, among which some are DNE and some are not. As shown in Figure 3, while GDA may converge to fixed points that are not local Nash equilibria, double-FTR avoids such spurious fixed points. Also, in the neighbourhood

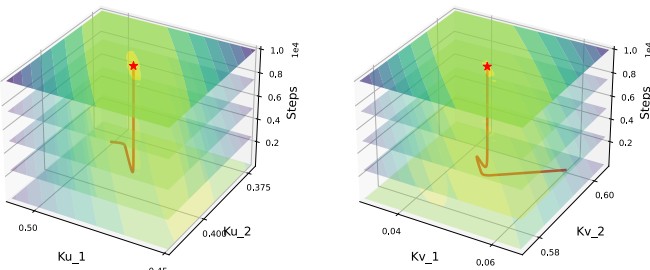

**Figure 4:** Evolution of the loss landscape of a general-sum linear quadratic game when optimized by double-FTR. We visualize two 2D slices $(K_{u,1}, K_{u,2})$ and $(K_{v,1}, K_{v,2})$ and the loss functions $f_u$ and $f_v$ respectively. As seen on the top levels of the illustration, the loss landscape is "bowl-shaped" at convergence, confirming that double-FTR solution satisfies the second-order conditions for DNE.

of local Nash equilibria, GDA exhibits oscillatory behaviour due to complex eigenvalues of the Jacobian matrix. In contrast, the double-FTR does not have oscillatory behaviour near local Nash equilibria. For reference, we also show the trajectories of the Local Symplectic Surgery (LSS). In this experiment, LSS has similar convergence properties – it avoids spurious fixed points and does not have oscillatory behaviour near local Nash equilibria.

## 6.2 GENERAL-SUM LINEAR QUADRATIC GAME

The linear quadratic (LQ) game is a classic problem in multi-agent learning. It is an extension of the famous linear quadratic regulator (LQR) problem of optimal control to the multi-agent setting. Just as LQR being a simple yet important benchmark problem for studying properties of reinforcement learning algorithms, the LQ game provides valuable insights to multi-agent RL algorithms (Fazel et al., 2018; Zhang et al., 2019a).

Consider the discrete-time linear dynamical system, where $\boldsymbol{z} \in \mathbb{R}^{d_z}$ is the state, and two players provide control inputs $\boldsymbol{u} \in \mathbb{R}^{d_u}$ and $\boldsymbol{v} \in \mathbb{R}^{d_v}$ respectively.

$$\boldsymbol{z}_{t+1} = \boldsymbol{A}\boldsymbol{z}_t + \boldsymbol{B}_u\boldsymbol{u}_t + \boldsymbol{B}_v\boldsymbol{v}_t, \quad \boldsymbol{z}_0 \sim p(\boldsymbol{z}_0)$$

Each player adopts a linear state-feedback policy: $\boldsymbol{u}_t = -\boldsymbol{K}_u\boldsymbol{z}_t$, $\boldsymbol{v}_t = -\boldsymbol{K}_v\boldsymbol{z}_t$, where the parameters $\boldsymbol{K}_u \in \mathbb{R}^{d_u \times d_z}$, $\boldsymbol{K}_v \in \mathbb{R}^{d_v \times d_z}$ are to be determined by optimization. In a general-sum LQ game, each player aims to find their corresponding policy parameters $\boldsymbol{K}$ that minimizes their individual quadratic loss function $f$ (shown in equation 4, $f_v(\boldsymbol{K}_u, \boldsymbol{K}_v)$ defined analogously using $\boldsymbol{Q}_v$ and $\boldsymbol{R}_v$).

$$f_u(\boldsymbol{K}_u, \boldsymbol{K}_v) = \mathbb{E}_{\boldsymbol{z}_0 \sim p(\boldsymbol{z}_0)}\left[\sum_{t=0}^{\infty} \boldsymbol{z}_t^\top \boldsymbol{Q}_u \boldsymbol{z}_t + \boldsymbol{u}_t^\top \boldsymbol{R}_u \boldsymbol{u}_t\right] \quad (\boldsymbol{Q}_u \succ 0, \ \boldsymbol{R}_u \succ 0) \tag{4}$$

Despite their simplicity, LQ games are challenging to optimize, because even though the loss functions are quadratic in the states and actions, they are *not* convex with respect to the player parameters $\boldsymbol{K}_u$ and $\boldsymbol{K}_v$. Importantly, Mazumdar et al. (2020a) show that in general sum LQ games, using naive gradient-play almost surely avoids some Nash equilibria.

We demonstrate in general-sum LQ games, double-FTR is able to find the Nash equilibria that are avoided by gradient-play. We use a setting mentioned in Mazumdar et al. (2020a), where $d_z = 2$, $d_u = d_v = 1$, $\boldsymbol{R}_u = \boldsymbol{R}_v = 0.01$, and

$$\boldsymbol{A} = \begin{bmatrix} 0.511 & 0.064 \\ 0.533 & 0.993 \end{bmatrix}, \boldsymbol{B}_u = \begin{bmatrix} 1 \\ 1 \end{bmatrix}, \boldsymbol{B}_v = \begin{bmatrix} 0 \\ 1 \end{bmatrix}, \boldsymbol{Q}_u = \begin{bmatrix} 0.01 & 0 \\ 0 & 1 \end{bmatrix}, \boldsymbol{Q}_v = \begin{bmatrix} 1 & 0 \\ 0 & 0.147 \end{bmatrix}.$$

The initial state $\boldsymbol{z}_0$ is set to $\begin{bmatrix} 1 & 1 \end{bmatrix}^\top$ or $\begin{bmatrix} 1 & 1.1 \end{bmatrix}^\top$ with equal probability.

Figure 4 and 5 shows an instance where the double-FTR is able to converge to a local Nash equilibrium, but the gradient-play fails to. For both algorithms, we use the same initial policy parameters $\boldsymbol{K}_u$ and $\boldsymbol{K}_v$. Figure 4 visualizes the loss landscape for $f_u(\boldsymbol{K}_u, \boldsymbol{K}_v)$ and $f_v(\boldsymbol{K}_u, \boldsymbol{K}_v)$ when optimized by double-FTR. It confirms that the solution double-FTR converges to is indeed a Nash equilibrium (the

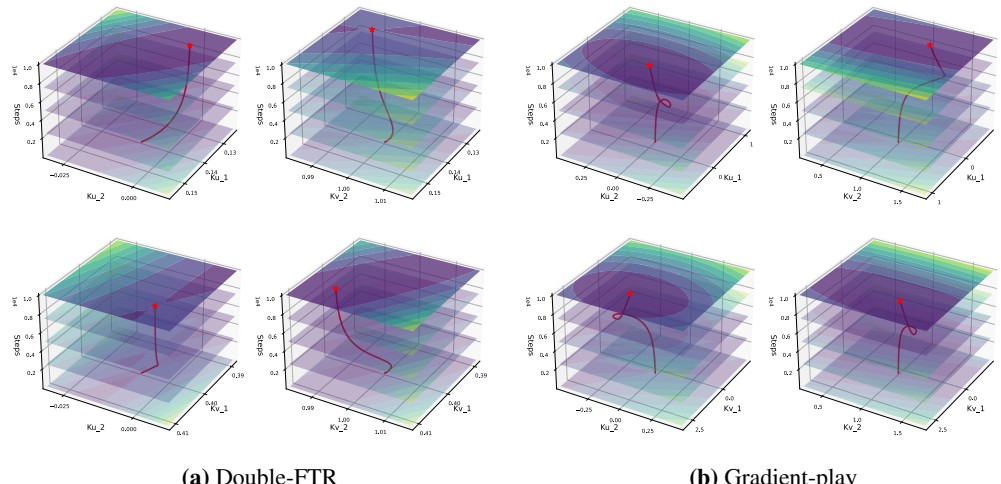

**(a)** Double-FTR        **(b)** Gradient-play

**Figure 5:** For the general-sum linear quadratic game, we visualize the evolution of the vector field Jacobian through different 2D slices. At each step, the contour plot visualizes the quadratic function defined by the current $J_{\text{GDA}}$, centered at the current weight values. In both (a) and (b), the weights are initialized near a DNE that is an *unstable* fixed point for GDA dynamics. (a): using double-FTR, the weights converge to the DNE, where $J_{\text{GDA}}$ has negative eigenvalues (shown as saddle-points on the contour maps). (b): the gradient method avoids this unstable DNE, and converges to a stable fixed point instead.

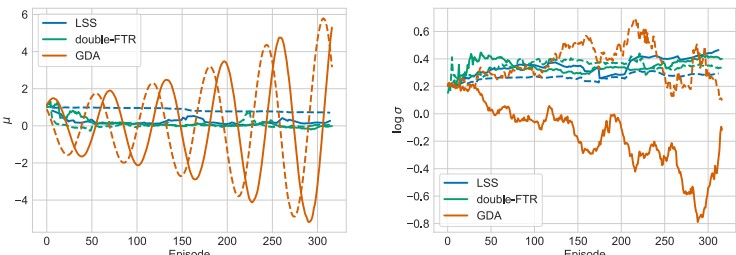

**Figure 6:** Weight evolution of the parameterized bilinear game under GDA and double-FTR. Solid line represent the weights for $\boldsymbol{\theta}$, and dashed lines represent the weights for $\boldsymbol{\phi}$.

second-order condition in Definition 2.1). Figure 5a visualizes the local vector field Jacobian (i.e. $J_{\text{GDA}}$) and shows that the Jacobian contains negative eigenvalues, which makes it a saddle point for gradient-play optimization. Indeed, gradient-play (shown in Figure 5b), avoids this Nash equilibrium. Instead, it eventually finds another Nash equilibrium that is stable fixed point.

### 6.3 PARAMETERIZED BILINEAR GAME

We consider another zero-sum game, the stochastic parameterized bilinear game, as in Prajapat et al. (2021). We use this experiment to demonstrate that double-FTR is also beneficial for stochastic games, and does not exhibit oscillatory behaviour due to having real eigenvalues at fixed points.

$$\min_{\mu_x, \sigma_x} r(x, y), \quad \min_{\mu_y, \sigma_y} -r(x, y) \quad \text{where } x \sim \mathcal{N}(\mu_x, \sigma_x^2), \ y \sim \mathcal{N}(\mu_y, \sigma_y^2), \ r(x, y) = xy.$$

The unique Nash equilibrium with respect to $(x, y)$ is $(0, 0)$. However, the learnable parameters are the mean and the standard deviation of the distributions where $x$ and $y$ are drawn from. We denote the learnable parameters for $x$ and $y$ as $\boldsymbol{\theta}$ and $\boldsymbol{\phi}$ respectively. At each time step, we obtain an unbiased estimate of the gradient using REINFORCE over a mini-batch of size $B$:

$$\tilde{\nabla}_{\boldsymbol{\theta}} r(\boldsymbol{\theta}, \boldsymbol{\phi}) = \frac{1}{B} \sum_{i=1}^{B} \nabla_{\boldsymbol{\theta}} \log \mathcal{N}(x_i; \boldsymbol{\theta}) r(x_i, y_i), \quad \boldsymbol{\theta} = \begin{bmatrix} \mu_x \\ \sigma_x \end{bmatrix}, \quad \tilde{\nabla}_{\boldsymbol{\phi}} r(\boldsymbol{\theta}, \boldsymbol{\phi}) \text{ computed analogously.}$$

As is often the case, GDA has oscillatory behaviour due to the complex eigenvalues of its Jacobian at fixed points. In this stochastic setting, the oscillation prevents convergence for GDA (Figure 6). In contrast, the double-FTR algorithm does not have rotational behaviour at fixed points, and converges to the unique Nash equilibrium $(x, y) = (0, 0)$.

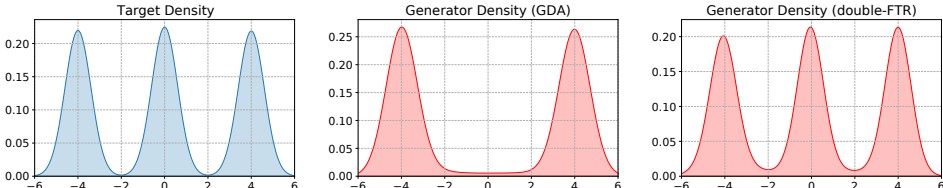

**Figure 7:** Mixture of Gaussians in 1D. Left: ground-truth. Middle: generator distribution learned by GDA. Right: generator distribution learned by double-FTR.

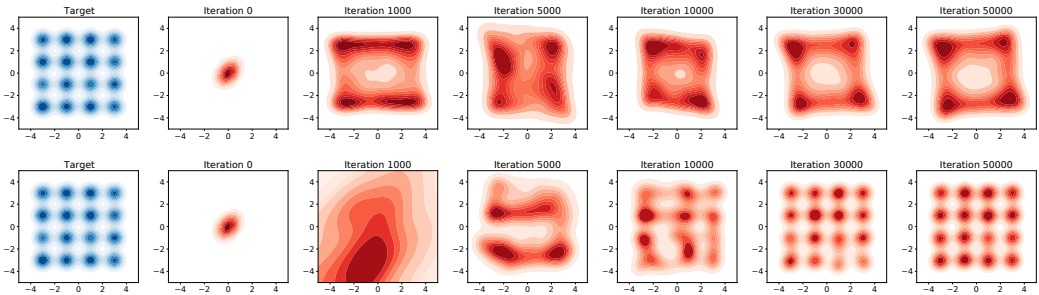

**Figure 8:** Mixture of Gaussians in 2D. Top: GDA suffers from mode collapse. Bottom: the generator distribution learned by double-FTR recovers all the modes.

### 6.4 GENERATIVE ADVERSARIAL NETWORKS

The Generative Adversarial Network (GAN) (Goodfellow et al., 2014) is a popular deep learning application for two-player games. The goal is to find the Nash equilibrium where the generator perfectly matches the target distribution, and the discriminator is completely fooled by the generator.

In this experiment, we use the GAN framework to learn mixture of Gaussians (MoG). We use the original saturating loss function. Both the generator and the discriminator are multi-layer perceptrons with 2 hidden layers and 64 hidden units in each layer. With neural networks, directly implementing the Hessian would be computationally inefficient or infeasible. Instead, we use conjugate gradient to approximate the Hessian inverse. Details of the experiments can be found in Appendix C.

As shown in Figure 7 and 8, we apply GDA and double-FTR to learn MoG in 1D and 2D. In both cases, GDA gets stuck at a spurious equilibrium and suffers from mode collapse. In contrast, double-FTR recovers all the modes, and the generated distribution closely matches the target.

## 7 CONCLUSION

We propose double Follow-the-Ridge (double-FTR), a gradient-based algorithm for finding local Nash equilibria in differentiable games. We prove that under mild assumptions, double-FTR locally converges to and only to differential Nash equilibria in the general-sum games, and avoids oscillation in the neighbourhood of fixed points. Furthermore, we remark that by varying the preconditioner, double-FTR leads to a broader family of algorithms that share the same convergence guarantee. Finally, we empirically verify the effectiveness of double-FTR in finding and only finding local Nash equilibria across a broad range of problems.

## 8 REPREDUCABILITY STATEMENT

For empirical results, we describe the experiment settings in detail in Appendix C. We also provide code for all experiments in the supplementary material. For the theoretical results, proofs are included in the appendix.

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
