# OpenReview forum: "Finding and only finding local Nash equilibria by both pretending to be a follower"
_ICLR.cc/2023/Conference — Submitted to ICLR 2023_

### Official Review · Reviewer_2AFw · 2022-10-22

**Confidence:** 3
**Correctness:** 4
**Technical Novelty And Significance:** 3
**Empirical Novelty And Significance:** 3
**Recommendation:** 5

**Clarity, Quality, Novelty And Reproducibility:**

The paper is clearly written and the result is well-presented. The main issue is the paper's limited technical novelty.

**Strength And Weaknesses:**

The main result of the paper is interesting and clearly presented. The proposed algorithm can be intuitively seen as an extension of the standard GDA algorithm where the update of each player has an additional term that makes use of second-order information. In particular, it seems like a natural extension of the FTR algorithm of [Wang, Zhang, Ba, ICLR 2019]. Having read the appendix, my main concern has to do with the technical novelty and challenges behind the main result. My concerns focus specifically on two aspects: (i) the algorithm is heavily similar to the FTR algorithm of [WZB '19] and (ii) the technical challenges (provided Assumption 2) are mainly algebraic manipulations of the double-FTR dynamics.

I would like to pose the following questions:

Q1. Can the authors highlight the technical novelties and challenges behind the result? Adding a technical overview section in the main body would be helpful.

Q2. Assumption 2 ensures that the updates in lines 2 and 3 of Algorithm 1 do not vanish. Is it possible to drop this assumption or relax it by extending the proposed algorithm?

Q3. Is it known whether second-order information is truly necessary in order to obtain a result like Theorem 1? As discussed in Section 2, standard GDA fails to converge even in zero-sum games. It would be beneficial to add a more extensive discussion regarding other first-order methods.

In general, I believe that the paper is nice and well-written. However, I think that the technical contribution of the paper is limited. Nevertheless, I am willing to increase my score after authors' rebuttal depending on the response.


**Summary Of The Paper:**

This paper studies general-sum two-player differentiable games and proposes an algorithm (double-FTR) that locally converges to (and only to) differentiable Nash equilibria (DNE). Moreover, extensive empirical validation is provided.


**Summary Of The Review:**

The paper proposes double-FTR algorithm which locally converges to and only to DNE. This algorithm is quite close to the FTR algorithm of [Wang, Zhang, Ba, ICLR 2019] and its analysis contains limited novelty. On the other side, the paper is well-written and the result is nice. This renders the paper marginally below the acceptance threshold.

---

> ### Author Response · Authors · 2022-11-14
> **Thank you very much for your review! Please find our responses below.**
>
> Thank you very much for your comments and good questions! Please find our answers below:
>
> **Q1:** The main technical novelties of our paper are: 1) showing that a simple algorithm (Algorithm 2) leads to local convergence results to and only to DNEs in general sum games; and 2) the proposed algorithm has elegant connections with several existing algorithms, such as FTR and the Hamiltonian Gradient Descent.
>
> With that said, we would like to emphasize that we are proposing a new algorithm with a first-of-its-kind convergence guarantee, and that the simplicity of our algorithm and proof techniques should not be considered a weakness of our paper. While complexity and advanced techniques are needed in many theoretical works, simple solutions should be welcomed as well. We believe that our novel results make meaningful contributions to the community.
>
> **Q2:** Thanks for the good question. Currently, we do need this assumption to ensure that the updates of Algorithm 2 do not vanish (note that a similar assumption is needed in [Mazumdar, 2019]. Dropping or relaxing this assumption would be deferred to future work.
>
> **Q3:** While we don’t have a solid proof on this issue, we do believe that some kind of second-order information would be necessary to obtain a result like Theorem 1.
> * Intuitively, GDA fails to converge even to some zero-sum games because in those zero-sum games, the Jacobian has a large anti-symmetric component, causing convergence to spurious fixed points. This also holds true for other first-order algorithms such as optimistic gradient descent (OGDA) and extragradient (EG).
>
> * For example, when $f(x,y) = g(x,y) = 2x^2 + 5xy + y^2$ (as in Figure 2(a)), the Jacobian of the gradient field $\begin{bmatrix}4 & 5; -5 & -2\end{bmatrix}$ has a large anti-symmetric component. Even though $g(x,y)$ is not concave w.r.t at $(0,0)$, the strong anti-symmetric component causes the eigenvalues to have a large imaginary component instead of a negative real part. It’s straightforward to show that in this case, first-order methods such as GDA, OGDA and EG all converge to the spurious fixed point at $(0, 0)$. We have added the OGDA and EG trajectories to Figure 2(a).
>
> * Since the analysis is based on the eigenspectrum of the Jacobian matrix, we believe that first-order methods without any means of estimating the Jacobian would not be able to avoid the spurious fixed points. This is also supported by the fact that so far no first-order (or else) algorithms have the convergence guarantee as in our Theorem 1.
>
>
>
> We hope that we have addressed your concerns and questions. We hope you would consider increasing your score if you find the answers satisfactory. If you would like to clarify or have further questions, we are eager to continue the discussion.
>
> ---
> **Reference:**
>
> Mazumdar, Eric V., Michael I. Jordan, and S. Shankar Sastry. "On finding local nash equilibria (and only local nash equilibria) in zero-sum games." arXiv preprint arXiv:1901.00838 (2019).

---

> > ### Comment · Reviewer_2AFw · 2022-11-17
> > **Thanks for the response.**
> >
> > I would like to thank the authors for the rebuttal and for their response to my questions.
> >
> > I believe that the paper is nice; however, I still think that the paper's technical novelty places this work slightly below the acceptance threshold. Hence, I would like to keep my score unchanged.

---

> > > ### Author Response · Authors · 2022-11-18
> > > **Thank you for your reply!**
> > >
> > > Thanks for your response.
> > >
> > > While we do believe that our contributions are significant (a novel algorithm with a first-of-its-kind convergence guarantee), and in which the technical novelty in the sense of derivation & proof techniques shouldn’t be considered a main part, we respect different opinions and appreciate your response.

---

### Official Review · Reviewer_tEUJ · 2022-10-24

**Confidence:** 4
**Clarity, Quality, Novelty And Reproducibility:** The paper is well written and well-mo…
**Correctness:** 4
**Technical Novelty And Significance:** 3
**Empirical Novelty And Significance:** Not applicable
**Recommendation:** 6

**Strength And Weaknesses:**

Pros:
    * the paper is self-contained and all the definitions are provided
    * the novel method is a natural extension of the Follow-the-Ridge

Cons:
    * the convergence of the algorithm is guaranteed only locally
    * although the paper contains a lot of preliminaries which is helpful, the main body contains a lot of facts that are standard in min-max optimization and could be moved to the appendix
    * a standard reference for the limits of gda is not contained in the paper, i.e.,

Daskalakis, C. and Panageas, I., 2018. The limit points of (optimistic) gradient descent in min-max optimization. Advances in neural information processing systems, 31.

**Summary Of The Paper:**

In this paper, the authors contribute a novel method, Double-Follow-the-Ridge, for solving the problem of computing local Nash equilibria in two-player general-sum differentiable games. This novel method is inspired by the Follow-the-Ridge heuristic algorithm and uses second-order information along with first-order information. In fact, the authors generalize the method to contain dynamics that use preconditioners in place of the second order information as long as some conditions are met. Finally, the paper is complemented with an array of experiments that demonstrate the empirical performance of the proposed method. The convergence guarantees are restricted to the local convergence regime.


**Summary Of The Review:**

The paper examines a well-established problem (min-max optimization of general-sum differentiable games), and it presents a novel method that builds upon existing work. The result is not particularly surprising although nontrivial. The idea of the novel method builds upon an existing algorithm but (probably justifiably) the authors did not move beyond local-convergence guarantees. As such, I believe the paper is slightly above the bar for acceptance.

---

> ### Author Response · Authors · 2022-11-14
> **Thank you very much for your review! Please find our responses below.**
>
> Thank you for your positive review and your helpful comments! Below are our responses to the raised issues / suggestions:
>
> **Only local convergence results are provided:** We focus on the local convergence results, because aiming for global convergence to and only to NE in general-sum games would be too ambitious (even determining if a game has a pure NE is NP-hard [Georg et al., 2005], and finding mixed Nash equilibria in general-sum games is PPAD-complete [Daskalakis et al., 2009]). For general-sum differentiable games, our algorithm is the first to have local convergence to and only to DNE.
>
> **Move some preliminaries to the appendix:** Thanks for suggesting this structural change. In the camera-ready version, we will focus on giving readers intuition in the preliminary part, and move some additional details to the appendix.
>
> **Missing reference for the limits of GDA:** Thanks for pointing out the missing reference -- we have added it to the updated version.
>
> We hope that we have addressed your concerns and questions. If you would like to clarify or have further questions, we are eager to continue the discussion.
>
> ---
> **References:**
>
> Gottlob, Georg, Gianluigi Greco, and Francesco Scarcello. "Pure Nash equilibria: Hard and easy games." Journal of Artificial Intelligence Research 24 (2005): 357-406.
>
> Daskalakis, Constantinos, Paul W. Goldberg, and Christos H. Papadimitriou. "The complexity of computing a Nash equilibrium." SIAM Journal on Computing 39.1 (2009): 195-259.

---

### Official Review · Reviewer_2z6c · 2022-10-27

**Confidence:** 5
**Correctness:** 3
**Technical Novelty And Significance:** 3
**Empirical Novelty And Significance:** 3
**Recommendation:** 5

**Clarity, Quality, Novelty And Reproducibility:**

Clarity & Quality: It is well-written paper
Novelty: Not-clear but interesting.

See above.

Clarification question #1: The preconditioning or the regularization with \ell_2 could transform the game to strong convex-concave or not?
Clarification question #2: In https://arxiv.org/abs/1910.13010, cited by (https://arxiv.org/pdf/2210.09769.pdf), difficult non-convex non-concave games are presented. It would be useful to present experiments of double FTR in such so-called hidden games, in order to exemplify the success of local vs global convergence guarantee in some challenging functions (since the presented ones seem in my humble opinion a bit artificial & synthetic)

**Details Of Ethics Concerns:**

Non-applicable

**Strength And Weaknesses:**

Strength:
The iead of introducing of being the doubly the follower is interesting. It is not surprising that replacing the opponents' strategy by a follower's one will lead to better and safest performance. On the other hand, it seems to be a simpler and more general way (like a blackbox) to doing lookahead than previous approaches by modeling opponents.

Weaknesses:
1)There is no theoretical guarantee on convergence or approximation bound.
2)The double Follow the ridge trick may be too simple. Anyway, there should be a comparison with simply lookahead methods and with the following publications:

a) https://arxiv.org/pdf/2210.09769.pdf
b) https://arxiv.org/abs/2112.13826
3) The method is second-order.

**Summary Of The Paper:**

This paper proposes the application of Follow-The-Ridge (usage of Shur's complement in Gradient Dynamics) for both players. The authors establish that double Follow-the-Ridge (double-FTR) is an algorithm that locally converges to and only to local Nash equilibria in general-sum two-player differentiable games.

**Summary Of The Review:**

The authors propose double Follow-the-Ridge (double-FTR), an algorithm with local convergence guarantee to differential Nash equilibria in general-sum two-player differential games.

---

> ### Author Response · Authors · 2022-11-14
> **Thank you very much for your review! Please find our responses below.**
>
> **Weaknesses:**
> 1) **Theoretical guarantee:** Thanks for raising this concern. However, we kindly point to our Theorem 1 & Corollary 1, which provide local convergence guarantee to and only to DNEs in general-sum differentiable games.
>
> 2) **Comparison to lookahead methods:** Thanks for suggesting the references to the lookahead methods. We would like to point out that these lookahead methods still do not have convergence guarantee only to Nash equilibria -- they might still converge to spurious, non-Nash fixed points. We will add these discussions to a revised version of the paper.
>
> 3) **Our method is second-order:** We believe that some kind of second-order information is necessary to obtain a result like Theorem 1. While we don’t have a solid proof on this issue, here is our reasoning:
> * Intuitively, GDA fails to converge even to some zero-sum games because in those zero-sum games, the Jacobian has a large anti-symmetric component, causing convergence to spurious fixed points. This also holds true for other first-order algorithms such as optimistic gradient descent (OGDA) and extragradient (EG).
>
> * For example, when $f(x,y) = g(x,y) = 2x^2 + 5xy + y^2$ (as in Figure 2(a)), the Jacobian of the gradient field $\begin{bmatrix}4 & 5; -5 & -2\end{bmatrix}$ has a large anti-symmetric component. Even though $g(x,y)$ is not concave w.r.t at $(0,0)$, the strong anti-symmetric component causes the eigenvalues to have a large imaginary component instead of a negative real part. It’s straightforward to show that in this case, first-order methods such as GDA, OGDA and EG all converge to the spurious fixed point at $(0, 0)$. We have added the OGDA and EG trajectories to Figure 2(a).
>
> * Since the analysis is based on the eigenspectrum of the Jacobian matrix, we believe that first-order methods without any means of estimating the Jacobian would not be able to avoid the spurious fixed points. This is also supported by the fact that so far no first-order (or else) algorithms have the convergence guarantee as in our Theorem 1.
>
> **Novelty:** we hope to further clarify our novelty by emphasizing that our method is the first to guarantee local convergence to and only to DNE in general-sum differentiable games.
>
> **Clarification questions:**
> 1. Our gradient correction terms (or equivalently, the preconditioning to the Jacobian) can be understood as locally transforming the game to strongly convex-strongly concave.
> 2. Thanks for the question. We’d like to mention that our experiment on parameterized bilinear game (section 6.3) is actually a kind of hidden game. Also, as opposed to the suggested papers (thanks for the reference), our focus is not on the structure of the game, but rather the ability to converge to DNE in the general case. We believe that our experiments involve games that are complex enough (non-convex concave, and involve a large number of parameters in the case of GAN) to demonstrate the effectiveness of our algorithm.
>
> We hope that we have addressed your concerns and questions. If you would like to clarify or have further questions, we are eager to continue the discussion.

---

### Official Review · Reviewer_PeZQ · 2022-10-27

**Confidence:** 4
**Clarity, Quality, Novelty And Reproducibility:** See above.
**Correctness:** 3
**Technical Novelty And Significance:** 4
**Empirical Novelty And Significance:** 2
**Recommendation:** 5

**Strength And Weaknesses:**

Strengths
======================
  - The problem that the authors consider is very important and difficult, and hence even the local convergence results in this area have significant theoretical importance.

  - The paper is well-written and the algorithms and techniques well-explained.

  - The dynamics that the authors provide are simple and intuitive and can potentially have significant practical applications.

Weaknesses - Comments
========================

  General comment: I believe that the results of this paper are indeed interesting but I think their current motivation is wrong. I will justify this with the comments that I present below.

  1. If we assume that there exists a Differential Nash Equilibrium according to Definition 2.1 and we also assume Assumption 1 then, unless I am missing something, f is convex with respect to x and g is concave with respect to y everywhere.
  Proof sketch: Let $z^* = (x^*, y^*)$ be a Differential Nash Equilibrium according to Definition 2.1. Assume also that there exist a point $z_0 = (x_0, y_0)$ such that $f$ is not convex with respect to $x$ at $z_0$. This means that $\nabla^2_{x x} f(z_0)$ has a negative eigenvalue. Consider now the line segment $L$ that connects $z^*$ and $z_0$. We know that the eigenvalues of $\nabla^2_{x x} f(z^*)$ are all positive and that $\nabla^2_{x x} f(z)$ is a continuous function of $z$ according to Assumption 1. It is also well-known that the eigenvalues are continuous functions of the matrices. Hence, the minimum eigenvalue of $\nabla^2_{x x} f(z)$ goes from positive at $z = z^*$ to negative at $z = z_0$ in a continuous way. Since $\nabla^2_{x x} f(z)$ is symmetric all the eigenvalues are real and hence there should be a point $\bar{z} \in L$ such that the minimum eigenvalue of $\nabla^2_{x x} f(\bar{z})$ is zero which means that $\nabla^2_{x x} f(\bar{z})$ is not invertible which violates Assumption 1. Therefore, f is convex with respect to x and similarly g is concave with respect to y. Am I missing something here?
  Given the above, it is not an achievement of the proposed algorithm that is it stable only around stationary points that are second-order Nash because only such stationary points exist. So the main motivation of the paper is not correct.

  2. The achievement of the proposed algorithm though is that it is stable around all the simultaneous stationary points of f with respect to x and g with respect to y. This to my knowledge is a not trivial property even when we assume that f is convex with respect to x and g is concave with respect to y. The reason that this is interesting is that this is applied in a general-sum setting. Finding such simultaneous stationary points is as difficult as finding fixed points of any continuous map!
  Proof sketch: Let $H : \mathbb{R}^n \to \mathbb{R}^n$ be a continuous map and let $f(x, y) = \|x - H(y)\|^2$ and $g(x, y) = - \|x - y\|^2$. Then it is easy to see that all the simultaneous stationary points of f with respect to x and g with respect to y, correspond exactly to the fixed points of $H$.
  Finding fixed points of continuous maps is more general that finding solutions of general non-monotone variational inequalities, which is more general than multi-agent general-sum concave games which have a ton of applications in game theory and economics. So the fact that the proposed algorithm is locally asymptotically stable only on those points is an interesting property. Nevertheless, it should be compared to the known results from the variational inequalities community or the computation of fixed points community which is missing from the current version of the paper.

**Summary Of The Paper:**

  In this paper, the authors consider the problem of designing methods with local convergence for finding second order Nash equilibria in nonconvex two-player general-sum games. Their main result is that, under some assumptions, a variation of the Follow-the-Ridge algorithm is locally asymptotically stable around any second-order Nash equilibria of the game and is not locally asymptotically stable in any other point.

**Summary Of The Review:**

  Based on my comments above, I believe that there are some interesting corollaries from the work that is presented in this paper but the paper needs significant changes to reflect that and the current motivation is not correct unless I am missing something. At the current state I cannot recommend acceptance.

---

> ### Author Response · Authors · 2022-11-14
> **Thank you very much for your review! Please find our responses below.**
>
> Thank you very much for your thoughtful comments! Please find our responses below:
>
> 1. Thank you for pointing this out! You’re absolutely right that the current version of Assumption 1 limits the class of games to be convex-concave everywhere, which may not be very interesting. Fortunately, there is a simple fix to this:
>
> * Intuitively, we can remove the invertibility assumption if we revise Algorithm 2 such that whenever a certain dimension of $(\nabla_{xx}^2 f(z))$ is degenerate, we remove the gradient correction term (the last terms in Algorithm 2) and resort to GDA in that dimension.
>
> * Mathematically, because $(\nabla_{xx}^2 f(z))$ is symmetric, there exist orthogonal matrix $Q$ such that $(\nabla_{xx}^2 f(z)) = Q\Lambda Q^T$, where $\Lambda = diag(\lambda_1,\dots,\lambda_n)$ is a diagonal matrix. Define $\bar{\Lambda}$ as:
>
>   * $$ \bar{\Lambda} := diag(\bar{\lambda}_1,\dots,\bar{\lambda}_n), ~~~\bar{\lambda}_i = \lambda_i^{-1} \text{ if } \lambda_i \neq 0 \text{ else } 0 $$
>   * The update for $x$ will instead be (analogously for $y$): $$x_{t+1} \leftarrow x_t - \eta_x \nabla_x f - \eta_y Q \bar{\Lambda} Q^T \nabla_{xy}^2 g \nabla_y g $$
>
>   With this simple fix, $\omega_{FTR}$ and $\omega_{GDA}$ still share the same fixed points, so the proof of Theorem 1 still holds.
>
> * Practically, instead of implementing the if statement, we can simply replace the $(\nabla_{xx}^2 f(z))^{-1}$ (in the additional gradient correction term) in Algorithm 2 with $(\nabla_{xx}^2 f \nabla_{xx}^2 f + \lambda I)^{-1} \nabla_{xx}^2 f$ with a small $\lambda > 0$, as already suggested at the end of Section 3 (and analogously for the y update). This way, it is almost identical with the original Algorithm 2 with small $\lambda$, but the update in the degenerate dimensions of $(\nabla_{xx}^2 f(z))$ is replaced with simple GDA.
>
> Thanks again for pointing this out. We have included the above discussion at the end of Section 3. While we would like to eventually fully remove the assumption on invertibility from Assumption 1, doing so would require iterating on the flow of presentation, because currently the $(\nabla_{xx}^2 f(z))^{-1}$ and $(\nabla_{yy}^2 g(z))^{-1}$ terms serve to give intuition for our algorithm. We will carefully revise this in the camera-ready version of the paper.
>
> 2. Thank you very much for pointing out the significance of our work in the computation of fixed points community! We will include a discussion in the related work section in a revised version of the paper.
>
> We hope that we have addressed your concerns and questions. If you would like to clarify or have further questions, we are eager to continue the discussion.

---

> > ### Comment · Reviewer_PeZQ · 2022-11-17
> > **Thank you for the response**
> >
> > 1. Overall it makes sense, although I believe it is a significant change to require a resubmission.
> >
> > 2. Observe that via fixed points the results can be applied to many problems in game theory and it would be nice if you can explore the implications of your work in some basic problems, e.g., computing Nash equilibrium in normal form games, or the general concave games setting of Rosen 1965.
> >
> > "Existence and uniqueness of equilibrium points for concave n-person games". JB Rosen. Econometrica 1965.

---

> > > ### Author Response · Authors · 2022-11-18
> > > **Thank you for your reply! Please find our additional responses below.**
> > >
> > > We suspect that we weren’t very clear in our reply to your first question, so please let us further clarify:
> > >
> > > In particular, we want to argue that the simple fix proposed above is not a significant change at all. Our reasons are as follows:
> > >
> > > **TL;DR: the “fix” is actually already included in our practical algorithm (including all our experiments), and the changes we need to make are mostly superficial (i.e. how to intuitively present our algorithm without mentioning the inverse).**
> > >
> > > * The current assumption on the invertibility of $(\nabla_{xx}^2 f(z))$ and $(\nabla_{yy}^2 g(z))$ is only there to ensure the naive version of Algorithm 2 makes sense. In practical implementation (including all our experiments), we never actually use the naive Algorithm 2. Instead, we make use of Remark 1 and use $(\nabla_{xx}^2 f \nabla_{xx}^2 f + \lambda I)^{-1} \nabla_{xx}^2 f$ instead of the naive inverse $(\nabla_{xx}^2 f(z))^{-1}$, and it no longer requires $(\nabla_{xx}^2 f(z))$ to be invertible.
> > > * The changes to be made to reflect this simple fix are primarily on the presentation of paper, as the purpose of currently including $(\nabla_{xx}^2 f(z))^{-1}$ and $(\nabla_{yy}^2 g(z))^{-1}$ in Algorithm 2 is for readers to intuitively understand our algorithm.
> > >
> > > Thank you again for your reply, and thanks for the suggestions on exploring the implications of our work on these other basic problems. However, we would appreciate it if you could re-evaluate our responses to your first question. Thank you very much!

---

### Decision · Program_Chairs · 2023-01-20

**Decision:**

Reject

**Justification For Why Not Higher Score:**

The authors treat an interesting problem, but given the earlier work of Mazumdar et al. are, the paper's contributions were not deemed sufficient and the change in the dynamics would require a thorough rewrite and a fresh set of reviews.

**Justification For Why Not Lower Score:**

N/A

**Metareview: Summary, Strengths And Weaknesses:**

This paper concerns the problem of finding Nash equilibria in two-player games where each player has a continuum of actions and smooth payoff functions. The authors' main contribution is a "double-FTR" (follow-the-ridge) policy with the following property: in two-player unconstrained games, a point is strictly stable under double-FTR if and only if it is a differential Nash equilibrium of the underlying game.

This contribution may be seen as an extension of an earlier result of Mazumdar et al. (2019) who showed that in two-player unconstrained *zero-sum* games, a point is stable under a certain preconditioned gradient process if and only if it is a differential Nash equilibrium. The method of the current paper is also a preconditioned gradient process, so the basic differences with the work of Mazumdar (2019) are that (a) the current paper treats general-sum games and (b) the dynamics have a simpler preconditioner.

Since the analysis hinges on the derivatives of the game's defining vector field near a solution, the zero-sum/general-sum extension was deemed relatively incremental compared to the work of Mazumdar et al. The simplification of the preconditioner is more interesting, but as was pointed out during the review phase, it comes at the cost of forcing the game to be convex-concave (because the individual Hessian of each player's loss function must maintain sign throughout, and it must be positive at the equilibrium in hand, so it must be positive everywhere). The authors's response to this point was to propose a change in the preconditioner; however, the proposed change brings the dynamics even closer to those studied by Mazumdar et al, so the proposed fix is somewhat tenuous and would require reviewing the paper from scratch.

There were also concerns that some of the authors' claims are not adequately supported by the theory. One has to do with the link between DNE and stable points of GDA: the authors state correctly that GDA could have stable points that are not DNE, not even Nash (this observation goes back at least to Daskalakis and Panageas, 2018). However, they also state the converse statement, namely that there exist DNE that are not stable under GDA - but the instability figures provided in the paper were not a clear indication of this.

Finally, the authors seem to suggest that double-FTR also avoids spurious limit cycles but, again, the evidence for this is not conclusive - in particular, it was not clear to what extent the avoidance illustrated in the experiments was initialization-dependent or not.

Because of the above limitations, a consensus was reached to make a "reject" recommendation.

**Summary Of Ac-Reviewer Meeting:**

Even though the scores of the paper were borderline, the committee quickly converged to a "reject" decision for the reasons outlined above.